# Characterization of *Gallibacterium anatis* Isolated from Pathological Processes in Domestic Mammals and Birds in the Czech Republic

**DOI:** 10.3390/pathogens13030237

**Published:** 2024-03-07

**Authors:** Jaroslav Bzdil, Soňa Šlosárková, Petr Fleischer, Monika Zouharová, Ján Matiašovic

**Affiliations:** 1Veterinary Research Institute, Hudcova 296/70, 621 00 Brno, Czech Republic; vetmed@seznam.cz (J.B.); sona.slosarkova@vri.cz (S.Š.); petr.fleischer@vri.cz (P.F.); monika.zouharova@vri.cz (M.Z.); 2Ptácy s.r.o., Valašská Bystrice 194, 756 27 Valašská Bystřice, Czech Republic

**Keywords:** cattle, poultry, clinical samples, prevalence, pathogenicity, genotyping, antimicrobial susceptibility

## Abstract

*Gallibacterium anatis*, recognized as a resident and opportunistic pathogen primarily in poultry, underwent investigation in unwell domestic mammals and birds. The study encompassed the mapping and comparison of *G. anatis* isolates, evaluation of their genetic diversity, and determination of their susceptibility to antimicrobials. A total of 11,908 clinical samples were analyzed using cultivation methods and MALDI-TOF. Whole-genome sequencing was performed on seven calf isolates and six hen isolates. Among mammals, *G. anatis* was exclusively detected in 22 young dairy calves, while among domestic birds, it was found in 35 individuals belonging to four species. Pathological observations in calves were predominantly localized in the digestive tract, whereas in birds, multi-organ infections and respiratory system infections were most prevalent. Distinct groups of genes were identified solely in calf isolates, and conversely, those unique to hen isolates were also recognized. Novel alleles in the multilocus sequence typing scheme genes and previously unidentified sequence types were observed in both calf and hen isolates. Antimicrobial susceptibility exhibited variation between bird and calf isolates. Notably, *G. anatis* isolates from calves exhibited disparities in genotype and phenotype compared to those from hens. Despite these distinctions, *G. anatis* isolates demonstrated the capability to induce septicemia in both species.

## 1. Introduction

*Gallibacterium anatis*, a member of the *Pasteurellaceae* family [1], is widely recognized as an inhabitant of the respiratory, intestinal, and genital tracts, playing an important role as an opportunistic pathogen in poultry. Its significance in poultry has recently been comprehensively reviewed [2,3]. While *G. anatis* is infrequently isolated and characterized as an opportunistic pathogen in mammals, particularly in cattle [1,4,5,6,7,8], it has also been found in a range of human cases including chronic bronchitis [9], lung abscess [10], bacteremia leading to death in an immunocompromised patient [11], and cases of diarrhea [12].

In early references, the tissue origin of *G. anatis* isolates in cattle was unknown [3,4,5]. However, recent studies have indicated that the isolates originate from the respiratory system [4,6,7]. Additionally, *G. anatis* has been detected in feces [5,8], and its presence has been noted in both cows [1,4] and calves aged 2–60 days [6,7,8]. With the exception of one isolate [1], all the studies mentioned in this paragraph described, if specified, various degrees of associated health disorders. Notably, none of these studies in cattle referred to septicemia or multi-organ infection caused by *G. anatis*.

In terms of pathogenicity, *G. anatis* possesses various factors that enable it to establish itself within the host organism [13]. These factors encompass the RTX-like toxin (GtxA), responsible for the microorganism’s hemolytic abilities, fimbriae facilitating adhesion to host cells, the formation of outer membrane vesicles (OMV), bacterial polysaccharide capsules, metalloproteases, and hemagglutinins, among others. Notably, *G. anatis* also demonstrates the ability to form a biofilm.

In a study conducted in Mexico, 23 isolates from chickens were examined, revealing a relatively broad spectrum of antimicrobial resistance [14]. All isolates (100%) exhibited resistance to penicillin, while 95.7% showed resistance to ampicillin, lincomycin, and tylosin. Additionally, 91.3% of isolates displayed resistance to oxytetracycline and enrofloxacin, and 87% were resistant to sulfamethoxazole and first-generation cephalosporins. Ceftiofur (26.1%) and florfenicol (34.8%) demonstrated the lowest levels of resistance. Similarly, other authors reported multiresistance in 58 avian isolates of *G. anatis*, with 65% showing multiresistance, and only 2 isolates being susceptible to all tested substances [15]. The most significant resistance was observed against tetracycline (92%) and sulfamethoxazole (97%).

The objective of this study was to map and compare the occurrence of *G. anatis* isolates in ill domestic mammals and birds in the Czech Republic. The study aimed to explore susceptibility to antimicrobial agents and investigate the genetic diversity of selected isolates using whole-genome sequencing. We also examined the genetic relatedness between our *G. anatis* isolates from chickens and calf diarrhea with recently published Belgian calf isolates from bronchopneumonia, aiming to gain insight into the *G. anatis* diversity.

## 2. Materials and Methods

### 2.1. Samples, Isolation and Identification of Bacteria

Sampling in vivo and examinations during autopsies were conducted as part of routine diagnostics. All samples were collected sterilely by 26 trained veterinarians and transported to the laboratory at temperatures ranging from +4 to +6 °C. Between 2013 and 2017, a total of 11,908 clinical and pathological samples from lesions and processes in clinically ill or dead domestic animals across 42 districts of the Czech Republic were analyzed. Of these, 7502 samples were clinical materials, while 4406 were pathological materials. Only one isolate per animal was included in the study. Of the samples, 10,743 originated from domestic mammals (including carnivores: 7661, cattle: 1668, rodents: 552, horses: 454, and pigs: 408), and 1165 came from domestic birds (including hens: 430, roosters: 35, chickens: 396, turkeys: 173, pigeons: 110, and ducks: 21). Except for chicken and calves, all other sampled animals were adults.

In cattle, the clinical materials comprised rectal swabs (n = 1104). The pathological materials (n = 564) were obtained from young dairy calves. These calves were typically housed in individual hutches or pens bedded with straw during the first weeks of life, and they were fed with a milk replacer and starter, following established protocols [8,16,17].

Clinical samples included rectal swabs, feces, swabs from the oral mucosa, hair, skin scrapings and swabs, urine, urinary tract swabs, respiratory tract swabs and lavages, pharyngeal swabs, conjunctival swabs, chest punctures, lymph node samples, and joint samples. During autopsies, samples from the heart, trachea, lungs, liver, spleen, and small intestine were collected for cultivation. In birds, air sacs were also examined when indicated. Sampling was conducted using Transbak system swabs with Amies agar and activated carbon, closable sterile plastic containers (60–200 mL capacity), sterile plastic tubes (10 mL capacity) with screw caps, or sterile plastic bags (Dispolab CZ s.r.o. Brno, Czech Republic).

In the laboratory, all samples were processed within 24 h of collection. Cultivation methods involved using meat peptone blood agar (MPBA) and Endo agar (EA) (both from Trios, s. r. o., Prague, Czech Republic), and plates were aerobically incubated at 37 ± 1 °C for 24–48 h. The growth of cultures was semi-quantitatively evaluated, and the results were reported as sporadic (+) to heavy (++++) growth [18]. Plates containing multiple bacterial cultures from organs lacking a microbiome were classified as contaminated without further identification. However, in plates with mixed bacterial cultures from organs or samples with a microbiome, the most frequently occurring colony-forming agent was considered the dominant culture, i.e., the dominant pathogen, and only these cultures were used in the study.

Suspected small grayish hemolytic or non-hemolytic colonies on MPBA plates were isolated, and subsequent pure cultures were identified using phenotypic molecular mass spectrometry, specifically matrix-assisted laser desorption/ionization coupled with time-of-flight mass spectrometry (MALDI-TOF). This process utilized a Microfex LT System spectrometer (Bruker Daltonik GmbH & Co. KG, Bremen, Germany), based on proteomics analyses, and MALDI Biotyper software MBT Compass 4.1 (Bruker Daltonik GmbH & Co. KG, Bremen, Germany) [19], with the MBT Compass Library Revision L 2020, covering 3239 species/entries (9607 MSP). Identification scores (ID) within the range of 2.300 to 3.000 were considered highly probable for species identification, 2.000 to 2.299 as secure genus identification and probable species identification, 1.700 to 1.999 as probable genus identification, and values ≤ 1.699 as unreliable identification (Bruker Daltonik GmbH & Co. KG, Bremen, Germany).

### 2.2. Whole Genome Sequencing (WGS)

Seven isolates of *G. anatis* from calves and five isolates from hens, along with the hen isolate CAPM 5995 from 1975 [20], were utilized for genomic comparison (see Appendix A). Genomic DNA was extracted from bacteria cultured on MPBA agar plates using the Qiagen DNeasy Blood and Tissue Kit (QIAGEN GmbH, Hilden, Germany). The DNA concentration of the samples was determined using a Qubit fluorometer (Thermo Fisher Scientific, Waltham, MA, USA). Subsequently, sequencing libraries were prepared with the Nextera XT DNA Sample Prep Kit (Illumina, San Diego, CA, USA) following the manufacturer’s protocol. Paired-end 2 × 150 bp sequencing was carried out using the NextSeq^®^ 500/550 High Output Kit v2 (Illumina, San Diego, CA, USA) and conducted on the NextSeq 500 instrument (Illumina, San Diego, CA, USA). Raw reads were deposited in the SRA under BioProject PRJNA1011493 (accessions SRR25865133–SRR25865145). Paired-end reads were analyzed using the TORMES 1.3.0 pipeline [21] in conjunction with the SPAdes assembler [22] with default settings. Species identification of assembled raw genomes was performed using Kraken2 [23] and 16S RNA analysis within the TORMES pipeline. Processing was parallelized using GNU Parallel [24]. Antimicrobial resistance (AMR) genes were identified by screening genomes against the Resfinder database [25] using Abricate (Seeman, https://github.com/tseemann/abricate (accessed on 2 December 2023)) [26] within the TORMES 1.3.0 pipeline.

### 2.3. Multi Locus Sequence Typing (MLST)

The mlst program (Seeman, https://github.com/tseemann/mlst (accessed on 2 December 2023)) [27] identified new allele sequences within the MLST scheme of *G. anatis* for all sequenced isolates. The sequences of these new alleles were submitted to the PubMLST database (https://pubmlst.org/ (submitted on 3 August 2023) [28], where they were assigned allele numbers, and subsequently, new sequence types were assigned to the isolates.

### 2.4. Comparison of Hen and Calf G. anatis Isolates

The draft genomes were annotated using Prokka [29] as part of the TORMES 1.3.0 pipeline. Raw reads of *G. anatis,* isolated from bronchopneumonia in calves [6], were downloaded from the SRA archive (https://www.ncbi.nlm.nih.gov/sra/?term=PRJNA541488; accessed on 2 December 2023) and assembled with the SPAdes assembler [22]. Then a neighbor-joining tree was constructed from the matrix based on the core-genes alignment created by Roary [30]. The tree was visualized using Mega [31]. The numbers within the nodes represent bootstrap values, while the length of the branches signifies the relative genetic distance. The genomic sequence CP002667.1 of the UMN179 strain [32], isolated from a hen, was used as the reference *G. anatis* sequence.

### 2.5. Antimicrobial Susceptibility Testing

Antimicrobial susceptibility testing was conducted using the disc diffusion method, following internationally recognized protocols established by the Clinical and Laboratory Standards Institute [33]. The Mueller–Hinton agar (Trios, s. r. o., Prague, Czech Republic) and antimicrobial discs from Oxoid Ltd. (Basingstoke, UK) were utilized. Tests were assessed after 18–24 h of incubation under aerobic conditions at 37 ± 1 °C. The tested antimicrobials, their concentrations, and interpretation criteria are summarized in Table 1. As there are no interpretative criteria for *G. anatis*, the categorization of isolates as susceptible, intermediate, and resistant was performed according to the *Pasteurellaceae* family. All used discs, media, and diagnostic methods underwent testing with reference strains of *Escherichia coli* (ATCC 25922), *Staphylococcus aureus* (ATCC 25923) (both obtained from the Czech Collection of Microorganisms, Masaryk University Brno, Czech Republic), and *G. anatis* (CAPM 5995) (Collection of Animal Pathogenic Microorganisms, Veterinary Research Institute Brno, Czech Republic).

## 3. Results

### 3.1. Prevalence

Out of the 11,908 clinical and pathological samples examined, isolates of *G. anatis* were obtained from 57 animals, resulting in a total prevalence of 0.5%. Semi-quantitative assessments in primary cultures revealed moderate (+++) or heavy (++++) growth. The prevalence of *G. anatis* in birds was 3.0%, while in mammals, it was 0.002%. Among mammals, *G. anatis* was exclusively isolated in cattle, with a prevalence of 1.3%, primarily in calves under 1 month old. Detailed data are presented in Table 2. Anamnestic data in this table indicate that the most common autopsy diagnosis in these animals was multi-organ infection, particularly prevalent in birds, accounting for 62.9% of birds with isolated G. anatis. In these cases, isolates were typically acquired from parenchymatous organs. Although only one calf had the autopsy diagnosis of multi-organ infection, *G. anatis* was isolated from the spleens of two other calves. The second most frequent diagnosis was gastroenteritis (along with suspected gastroenteritis; suspected due to diarrhea) in young calves (n = 20) and enteritis in birds (n = 8). The respiratory tract was the exclusive source of *G. anatis* isolates in 12 animals (only one of which was a calf). Pneumonia was recorded in 10 animals, aerosaculitis in 6 birds, tracheitis in 4 birds, and sinusitis in 1 duck. Additionally, myocarditis in 1 hen and nephritis in 1 rooster were reported. Table 2 provides the precise data for these diagnoses.

### 3.2. Genotyping

#### 3.2.1. Whole Genome Sequencing

Using Kraken2 and 16S rRNA analysis, all sequenced isolates were identified as *G. anatis*. A total of 1231 core genes were found in 99–100% of isolates, with an additional 1524 genes identified in 15–95% of isolates.

#### 3.2.2. AMR Genotyping

Using the ResFinder database, 14 different AMR genes conferring resistance to aminoglycosides, sulfonamides, tetracyclines, beta-lactams, phenicols, and trimethoprim were identified in *G. anatis* (Table 3).

Among hen isolates, a low occurrence of AMR genes was observed. In four out of six isolates, no AMR genes were detected. In the remaining two isolates, the genes *aph(3″)-Ib* and *aph(6)-Id*, conferring resistance to aminoglycosides, and the gene *sul2*, indicating resistance to sulfonamides, were identified (Table 3).

Calf isolates displayed a significantly higher incidence of AMR genes compared to hen isolates. All strains isolated from calves with diarrhea carried a minimum of two AMR genes. The most prevalent genes were those conferring resistance to tetracyclines, notably *tet(B)* in all seven isolates and *tet(M)* in five isolates, as well as aminoglycoside phosphotransferase genes (*aph*) in six isolates, which impart resistance to aminoglycosides. Three isolates harbored AMR genes for resistance to sulfonamides and phenicols. Additionally, two isolates carried an AMR gene targeting beta-lactamase-susceptible penicillins (*bla_CARB_*, *bla_ROB_*). In one isolate, a simultaneous identification of up to 10 AMR genes occurred.

Comparing our isolates to published results of Belgian strains isolated from calf bronchopneumonia [6], evident differences emerged (Table 3). Our hen isolates displayed a low average occurrence of AMR genes (1 per isolate), while calf isolates in our study showed a higher average (5 per isolate, up to 10 in one isolate). Belgian calf isolates from bronchopneumonia exhibited an even higher average (12 per isolate), with up to 14 AMR genes in one isolate. Although AMR genes in Belgian isolates targeted the same antibiotic substances, with the exception of macrolides (all Belgian isolates harbored the *ermB* gene, whereas none in our study did), the resistance was often encoded by different genes. These differences were most apparent for aminoglycosides, where resistance in our isolates was encoded by four genes, while in Belgian isolates, resistance was encoded by seven genes, of which only one (*aph(3′)-III*) was common to both groups of isolates. A similar finding was observed for beta-lactams, with resistance encoded by the *bla_ROB-1_* and *bla_CARB-16_* genes in our study, while the Belgian isolates carried the *bla_TEM-2_* and *bla_CARB-8_* genes.

#### 3.2.3. Multilocus Sequence Typing

All isolates exhibited new alleles in the MLST genes, specifically adk-27 to adk-29, atpD-28 to atpD-32, fumC-47 to fumC-52, gyrB-49 to gyrB-54, infB-38 to infB-44, mdh-30 to mdh-36, recN-38 to recN-45, and thdF-31 to thdF-35. Each isolate was associated with a unique sequence type, as detailed in Appendix A.

#### 3.2.4. Comparison of Hen and Calf Isolates

The analysis of genes present/absent in hen or calf isolates, respectively, revealed 13 genes uniquely present in all hen isolates but absent in all calf isolates (Appendix A). Similarly, 23 genes were found only in calf isolates (Appendix A). Four of the genes exclusively present in hen isolates are involved in fimbriae synthesis.

The core genome alignment of our calf and hen isolates, alongside Belgian calf isolates from bronchopneumonia [6], unveiled a mutual relationship among them. All hen isolates (H7, H8, H9, H10, G10, G11) and the reference CP002667.1 sequence formed a distinct cluster (Figure 1). With the exception of the isolate G9, all our calf isolates grouped together with one Belgian isolate from calf bronchopneumonia, creating a distinct cluster. Our isolate G9 and all remaining Belgian isolates from calf bronchopneumonia formed three additional distinct clusters.

#### 3.2.5. Susceptibility to Antimicrobials

Forty-three isolates underwent antimicrobial susceptibility testing using the disc diffusion method. Among the eight antimicrobial substances employed, the isolates exhibited the highest susceptibility to gentamicin (100%), amoxicillin/clavulanic acid (93%), penicillin G (92.7%), and colistin (90.5%). In contrast, the lowest susceptibility was observed with tetracycline (16.3%), enrofloxacin (41.9%), and co-trimoxazole (50%). Laboratory tests also revealed some differences in the susceptibility of *G. anatis* isolates obtained from calves and birds. In both groups of animals, 100% susceptibility was recorded only for gentamicin. In calves, 100% of isolates were susceptible to amoxicillin with clavulanic acid and in birds 90.9%.

Similar trends were observed for penicillin G, with 100% sensitivity in calves and 90.3% in birds. A high percentage of strains were susceptible to colistin (88.9% in calves and 90.9% in birds). On the other hand, a lower percentage of susceptible isolates were recorded in the case of sulfamethoxazole with trimethoprim, i.e., co-trimoxazole (44.4% sensitive isolates in calves and 55.6% in birds), enrofloxacin (10% sensitive in calves and 51.5% in birds), and also tetracycline (0% sensitive isolates in calves and 21.2% in birds). Exact data are presented in Table 4.

## 4. Discussion

The presented data confirm that the presence of *G. anatis* extends beyond avian species, suggesting its potential role as an opportunistic pathogen, albeit infrequently, in dairy calves. Notably, occurrences of *G. anatis* have also been documented in immunocompromised human patients, leading to bacteremia, respiratory tract disease, and diarrhea [9,10,11,12]. In addition to poultry, cattle thus also represent a potential reservoir for the zoonotic transmission of *G. anatis*. Further investigations are warranted to ascertain its zoonotic potential [6].

Although a diverse array of diseased domestic mammals were surveyed, encompassing a substantial number of samples (n = 10,743), particularly from carnivores (n = 7661), *G. anatis* was solely detected in cattle, specifically in young calves (up to weaning) from dairy farms. Czech dairy farms, relatively expansive within the broader European context [17], did not harbor poultry, similar to the conditions observed on Belgian farms as described by Van Driessche et al. [6]. The likelihood of calves being exposed to raw hen eggs was also deemed highly improbable. These intensively reared calves had minimal contact with birds compared to other mammalian groups in our study.

An analysis of the anamnestic data for our samples revealed that pathological processes and lesions in calves were primarily localized in the digestive tract in 20 cases, with pneumonia detected in only one case, and multi-organ infection recorded in another. In instances of pneumonia, multi-organ infection, and one case of gastroenteritis, *G. anatis* was also isolated from the spleen, suggestive of septicemia. Such occurrences in cattle have not been previously described in the literature. In birds, multi-organ infection was the most common diagnosis (22 cases), followed by respiratory system diseases (11 cases), with isolated incidents of myocarditis and nephritis. Unlike findings in the literature [1,13], no cases of salpingitis or peritonitis were observed in birds.

Susceptibility tests revealed a high resistance of the isolates to tetracyclines and potentiated sulfonamides, consistent with findings by other researchers [14,15]. Interestingly, isolates from birds exhibited differing susceptibility patterns compared to mammalian isolates, particularly for tetracycline and enrofloxacin. Screening *G. anatis* genomes for antimicrobial resistance (AMR) genes also unveiled disparities depending on the strain’s origin, with noticeably lower occurrences of AMR genes in hen isolates compared to calf isolates, indicating separate evolutionary trajectories for both populations. Notable differences were also observed between calf intestinal strains in our study and those isolated from bronchopneumonia in Belgium, where distinct AMR genes were identified. The divergent genetic basis for resistance to specific antimicrobials supports the notion of distinct phylogenetic lineages for both calf isolate groups, suggesting the existence of two separate *G. anatis* populations in cattle.

Our results from multilocus sequence typing (MLST) and core genome alignment further support the hypothesis of different *G. anatis* populations in birds and calves. The core genome alignment indicated a clear separation between the reference hen strain and all hen isolates analyzed, forming a distinct cluster apart from calf isolates. Notably, calf bronchopneumonia isolates from Belgium displayed greater genetic diversity compared to isolates from the digestive tract in the Czech Republic. While most digestive tract isolates and one bronchopneumonia isolate formed a single cluster, the remaining bronchopneumonia isolates segregated into three distinct clusters, despite originating from the same geographical region (Belgium). This divergence in clusters among calf isolates suggests a broader genetic variability compared to our hen isolates. However, it is worth noting that our analysis included only a small number of hen isolates from a single geographic area. To obtain deeper insights into the genetic diversity of *G. anatis* and the potential host specificity of different lineages, future studies should analyze a larger number of isolates from various regions worldwide.

The varied susceptibilities of avian and mammalian isolates, coupled with their genetic divergence, indicate distinct populations of *G. anatis*. Although the number of sequenced isolates is limited, the absence of transmission between birds and calves suggests that the source of *G. anatis* infection in calves likely originates from within the farms. This conclusion is consistent with the findings of Van Driessche et al. [6], who, among other factors, attributed the high genetic variability between isolates to the absence of a single introduction or outbreak, suggesting the presence of a large unsampled reservoir of circulating *G. anatis* strains in Belgian cattle. This idea extends beyond Belgium and Europe, finding support not only in our results but also in two recent studies from the USA, which found *Gallibacterium*, including *G. anatis*, present in varying degrees in the microbiome of newborn dairy calves fed milk replacer [8,16]. While bacteria of the genus *Gallibacterium* were primarily enriched during dysbiosis associated with diarrhea, *G. anatis* was also found in a group of control newborn calves with normal health status development, experiencing only mild and short-term diarrhea [8].

## 5. Conclusions

We uncovered that *Gallibacterium anatis* poses a potential threat to young dairy calves, being involved not only in gastroenteritis, diarrhea, and pneumonia but also septicemia, a phenomenon hitherto unreported in the literature. Additionally, apart from the genetic distance between mammalian and avian isolates, we observed differences in their affinity for specific organs and organ systems, as well as variations in sensitivity to antimicrobial substances.

## Figures and Tables

**Figure 1 pathogens-13-00237-f001:**
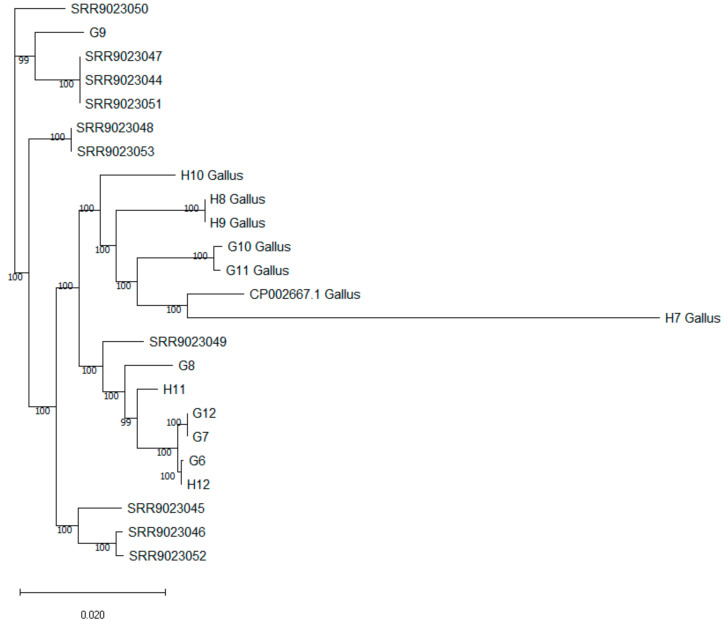
Phylogenetic tree based on core genome alignment. H7, H8, H9, H10, G10, G11, and the reference CP002667.1 are hen isolates; G6, G7, G8, G9, G12, H11, and H12 are isolates from calf diarrhea; SRR9023045-SRR9023053 are Belgian isolates from calf bronchopneumonia [6]. The numbers represent bootstrap values.

**Table 1 pathogens-13-00237-t001:** Antimicrobial susceptibility—reference values of inhibition zones for *Gallibacterium anatis*.

Antimicrobials	Antibiotics Concentration per Disc in µg (and IU)	Zone Diameter (mm)		
		R	S	Source
penicillin G (*Pasteurellaceae*)	6 (10)	<29	≥29	CASFM VET [34]
amoxicillin/clavulanic acid (*Pasteurellaceae*)	20/10	<14	≥21	CASFM VET [34]
cefalexin (*Pasteurellaceae*)	30	<12	≥18	CASFM VET [34]
trimethoprim/sulfamethoxazole (*Pasteurellaceae*)	1.25/23.75	<10	≥16	CASFM VET [34]
gentamicin (*A. pleuropneumoniae*)	10	≤12	≥16	CLSI VET [33]
tetracycline (*P. multocida*)	30	<24	≥24	EUCAST [35]
enrofloxacin (*Pasteurelaceae*)	5	<17	≥22	CASFM VET [34]
colistin (*Pasteurelaceae*)	50	<15	≥15	CASFM VET [34]

R = resistant; S = susceptible.

**Table 2 pathogens-13-00237-t002:** Number and origin of *Gallibacterium anatis* isolates from veterinary materials in the period 2013–2017.

Animal	Number of Animals/Samples	Number of Isolates (Prevalence %)	Origin of Isolates									Diagnosis
			Heart	Trachea	Lung	Air Sacs	Liver	Spleen	Kidney	Small Intestine	Rectal Swab, Feces	
Hen (adult)	430	18 (4.2)	12	15	15	15	11	13	12	7	0	11× multi-organ infection, 7× enteritis, 4× pneumonia, 4× aerosaculitis, 2× tracheitis, 1× myocarditis
Rooster (adult)	35	4 (11.4)	3	3	3	3	4	4	4	1	0	3× multi-organ infection, 1× nephritis, 1× enteritis
Chicken	396	3 (0.8)	2	3	3	2	2	3	2	1	0	2× multi-organ infection, 1× tracheitis, 1× pneumonia
Turkey (adult)	173	2 (1.2)	1	2	2	2	1	1	1	1	0	1× multi-organ infection, 1× pneumonia, 1× aerosaculitis
Pigeon (adult)	110	6 (5.5)	5	6	6	6	5	5	5	1	0	5× multi-organ infection, 1× pneumonia, 1× aerosaculitis
Duck (adult)	21	2 (9.5)	0	2	2	0	0	0	0	0	0	2× pneumonia, 1× tracheitis, 1× sinusitis
Cattle	1668	22 (1.3)	1	2	2	0	1	3	1	5	15	15× diarrhea, 5× gastroenteritis, 1× multi-organ infection, 1× pneumonia

Total	2833	57 (2.0)	24	33	33	28	24	29	25	16	15	23× multi-organ infection, 15× diarrhea, 10× pneumonia, 8× enteritis, 6× aerosaculitis, 5× gastroenteritis 4× tracheitis, 1× myocarditis, 1× nephritis, 1× sinusitis

**Table 3 pathogens-13-00237-t003:** AMR genes occurrence among 3 groups of *Gallibacterium anatis* isolates (hens and calves with diarrhea; calves with bronchopneumonia in the Belgian study conducted by Van Driessche et al. [6]).

		Aminoglycosides										Sulfonamides		Tetracyclines			Beta-Lactams			Trimethoprim	Phenicols			Macrolides	
Origin	Isolate	aph(3″)-Ib	aph(3′)-III	aph(6)-Id	aph(3′)-Ia	aadA	aadB	aphA1	strA	strB	aac(6)-aph(2)	sul1	sul2	tet(B)	tet(M)	other	blaROB	blaCARB	blaTEM	dfrA1	catA1	catA3	floR	ermB	Other
Czech hens	H10																								
	H7																								
	H8																								
	H9																								
	G10	aph(3″)-Ib		aph(6)-Id									sul2												
	G11	aph(3″)-Ib		aph(6)-Id									sul2												
Czech calves with diarrhea	G12		aph(3′)-III											tet(B)	tet(M)		blaROB-1								
	G6		aph(3′)-III											tet(B)	tet(M)										
	G7	aph(3″)-Ib	aph(3′)-III	aph(6)-Id									sul2	tet(B)	tet(M)	tet(H)		blaCARB-16		dfrA1			floR		
	G8													tet(B)	tet(M)										
	G9	aph(3″)-Ib											sul2	tet(B)						dfrA1	catA1	catA3			
	H11	aph(3″)-Ib		aph(6)-Id	aph(3′)-Ia								sul2	tet(B)								catA3			
	H12		aph(3′)-III											tet(B)	tet(M)										
Belgian calves with bronchopneumonia [6]	GB2					aadA1	aadB	aphA1	strA	strB			sul2		tet(M)						catA1	catA3	floR	ermB	
	GB3					aadA1	aadB	aphA1	strA	strB		sul1	sul2	tet(B)	tet(M)	tet(Y)		blaCARB-8	blaTEM-2				floR	ermB	
	GB4		aph(3′)-III			aadA1			strA		aac(6)-aph(2)		sul2	tet(B)	tet(M)				blaTEM-2	dfrA1	catA1			ermB	
	GB5					aadA1	aadB	aphA1	strA				sul2	tet(B)	tet(M)					dfrA1	catA1		floR	ermB	
	GB6					aadA1		aphA1	strA	strB		sul1	sul2	tet(B)	tet(M)	tet(Y)		blaCARB-8	blaTEM-2	dfrA1			floR	ermB	
	GB7					aadA1	aadB	aphA1	strA	strB			sul2	tet(B)	tet(M)				blaTEM-2		catA1	catA3		ermB	
	GB8					aadA23	aadB	aphA1	strA				sul2	tet(B)	tet(M)				blaTEM-2	dfrA1	catA1	catA3		ermB	mphE, mrsE
	GB9		aph(3′)-III			aadA1			strA		aac(6)-aph(2)		sul2	tet(B)	tet(M)				blaTEM-2	dfrA1	catA1			ermB	
	GB10					aadA1	aadB	aphA1	strA				sul2	tet(B)	tet(M)						catA1		floR	ermB	
	GB11		aph(3′)-III			aadA1			strA		aac(6)-aph(2)		sul2	tet(B)	tet(M)				blaTEM-2	dfrA1	catA1			ermB	

**Table 4 pathogens-13-00237-t004:** Numbers of susceptible and tested *Gallibacterium anatis* isolates and percentages of susceptible isolates.

	Number of Susceptible/Number of Tested Isolates Susceptible (%)						
Animal	Tetracycline	Penicillin G	Gentamicin	Enrofloxacin	Amoxicillin/ clavulanic acid	Co-Trimoxazole	Colistin
Birds	7/33 (21.2)	28/31 (90.3)	33/33 (100)	17/33 (51.5)	30/33 (90.9)	5/9 (55.6)	30/33 (90.9)
Mammals (calves)	0/10 (0)	10/10 (100)	10/10 (100)	1/10 (10)	10/10 (100)	4/9 (44.4)	8/9 (88.9)
Total	7/43 (16.3)	38/41 (92.7)	43/43 (100)	18/43 (41.9)	40/43 (93)	9/18 (50)	38/42 (90.5)

## Data Availability

The raw reads of sequenced isolates were deposited into SRA under BioProject PRJNA1011493 (https://www.ncbi.nlm.nih.gov/bioproject/?term=PRJNA1011493 (registration date 1 September 2023)). Raw data supporting the conclusions of this study are available from the authors upon request.

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
