# Peer review of "Characterization of Gallibacterium anatis Isolated from Pathological Processes in Domestic Mammals and Birds in the Czech Republic"

_pathogens, 2024, doi:10.3390/pathogens13030237_

Round 1

Reviewer 1 Report

Comments and Suggestions for Authors

Dear Authors,

The manuscript submitted for evaluation raises an interesting topic and certainly fits into current research trends. Recently, an increasing number of isolations of bacteria of the species Gallibacterium anatis have been observed. This microorganism is isolated primarily from commercial laying hens and from meat breeding hens. However, this bacterium can also be found in flocks of turkeys and broiler chickens. A new insight into the problem of G. anatis infections is certainly the analysis of the occurrence of these bacteria in cattle. The layout of the manuscript is correct. However, I have a lot of comments about the individual sections. A big problem is the construction of sentences, sometimes introducing ambiguity as to the content. Therefore, I believe that the work should be linguistically proofread. I am sending detailed comments below.

Detailed comments

Title

-I propose” “Characterization of Gallibacterium anatis isolated from pathological processes in domestic mammals and birds in the Czech Republic”

Abstract

-The sentence in lines 9-10 should be reworded to make it clear that the occurrence of G. anatis was analysed in domestic mammals and birds.

-Lines 12-13, "A total of 11,908 samples...". It is unclear whether 11,908 samples or 11,908 bacterial isolates were tested.

-Line 15, I suggest correcting: “…and among domestic birds in 35 individuals belonging to four species”

Introduction

-In the first paragraph, each sentence begins with "Gallibacterium anatis". I propose to reword this paragraph. Combine some sentences.

-Line 28, which means "relatively important"

-The sentence in lines 35-37 should be reworded.

-Line 37. I don't understand the statement "Regarding the known animal category or age of the calves."

-The sentence in lines 38-4 is difficult to understand. Perhaps the authors should take advice from a native speaker.

-Line 47, biofilms or biofilm?

Materials and Methods

Section 2.1. Samples, isolation and identification of bacteria

-I suggest rewording. First, specify which animals (species, age) were sampled and how many samples were collected. Time range, place. What types of samples were taken from living and dead animals. The total sum of samples taken and the total sum of animals from which samples were taken (individual animal species). Then the sampling technique and bacterial identification.

-Regarding samples taken from birds. The authors detailed how many samples were taken from hen, rooster and chicken. However, for turkeys, pigeons and ducks, age and gender were not taken into account. I suggest standardizing this data.

-Lines 66-67, instead of "Throughout the observed period..." I suggest: "In the period from 2013 to 2017..."

-Line 69, Each sample represents one animal. What does it mean?

-Sentence, line 73. All types of samples were evaluated in cattle. It is not clear.

-Line 74. The statement: Approximately 70% of these samples ... Is imprecise.

-Line 78, Almost no dairy calf received mature milk from its own mother. This information is imprecise.

-Lines 79-80, this sentence is redundant.

Section 2.2. Whole genome sequencing (WGS)

-Lines 117-118, Sentence: "Seven G. anatis isolates from calves and five of our isolates from hens, along with one reference isolate from a hen, were used for genomic comparison (Supplement 1)". If five "our" isolates from chickens were used, whose isolates were from calves?

-Table 1 is unnecessary since the authors cite the source in the methodology.

Results

-Line 177, "...prevalence within animal species" is redundant. All you need is the percentage in brackets.

-Lines 180-181, what does ".."categories of domestic chicken" mean?

-Lines 186-187. I don't understand this sentence: "While there was only one calf with the autopsy diagnosis of multi-organ infection, G. anatis was isolated from the spleen of two other calves."

-Lines 188-189. The sentence fragment in brackets is unnecessary.

-Lines 190-192. The sentence needs to be reworded because it is difficult to understand.

-Line 193. (Table 2) show the exact data for diagnoses.

-Table 2. Number and origin of Gallibacterium anatis isolates from veterinary materials in the period 194

2013 - 2017.

-Table 2 presents data for hens and roosters, and for ducks, turkeys and pigeons without gender division.

-In table 2. Diagnosis rubric - for calves is linguistic error: "eddiarrhea"

Section 3.2.2. AMR genotyping

-Line 207, please remove "at all"

-Lines 209-210, "In our calf isolates, the occurrence of AMR genes was notably higher" Notably higher than where?

-Paragraph 217-232 should be included in the discussion.

-Section 3.2.3. Comparison of hen and calf isolates

-Paragraph 246-252 should be included in the discussion.

-Section 3.2.4. Susceptibility to antimicrobials

-Line 260, the isolated isolates strains

-Line 266, what does "Good results" mean?

-Line 271, what does "weak susceptibility" mean?

-Line 274, (Table 4) shows the exact data.

Discussion

-Lines 291-299. These sentences contain elements of speculation. Information about nutrition and housing conditions, possible contacts with other animal species can be obtained from anamnesis.

-Lines 311-314, "The differences in diagnoses between calves and birds may be partly attributed to the numbers of sample types" Please rephrase the sentence because it is difficult to understand.

-Lines 321-322, I suggest rewriting the sentence: “Screening G. anatis genomes for AMR genes also revealed differences between groups of strains” to: “Screening G. anatis genomes for AMR genes also revealed differences depending on the origin of the strains.”

Conclusions

-Lines 371-373. These sentences are not conclusions. this is a repetition of the purpose and design of the study.

Comments on the Quality of English Language

A big problem is the construction of sentences, sometimes introducing ambiguity as to the content. Therefore, I believe that the work should be linguistically proofread. 

Author Response

Reviewer 1

Dear Authors,

The manuscript submitted for evaluation raises an interesting topic and certainly fits into current research trends. Recently, an increasing number of isolations of bacteria of the species Gallibacterium anatis have been observed. This microorganism is isolated primarily from commercial laying hens and from meat breeding hens. However, this bacterium can also be found in flocks of turkeys and broiler chickens. A new insight into the problem of G. anatis infections is certainly the analysis of the occurrence of these bacteria in cattle. The layout of the manuscript is correct. However, I have a lot of comments about the individual sections. A big problem is the construction of sentences, sometimes introducing ambiguity as to the content. Therefore, I believe that the work should be linguistically proofread. I am sending detailed comments below.

Detailed comments

Title

-I propose” “Characterization of Gallibacterium anatis isolated from pathological processes in domestic mammals and birds in the Czech Republic”

Response: Thank you for this suggestion. The title has been modified.

Abstract

-The sentence in lines 9-10 should be reworded to make it clear that the occurrence of G. anatis was analysed in domestic mammals and birds.

-Lines 12-13, "A total of 11,908 samples...". It is unclear whether 11,908 samples or 11,908 bacterial isolates were tested.

-Line 15, I suggest correcting: “…and among domestic birds in 35 individuals belonging to four species”

Response: Thank you for your suggestions. The Abstract has been rewritten with respect to your suggestions.

Introduction

-In the first paragraph, each sentence begins with "Gallibacterium anatis". I propose to reword this paragraph. Combine some sentences.

-Line 28, which means "relatively important"

-The sentence in lines 35-37 should be reworded.

-Line 37. I don't understand the statement "Regarding the known animal category or age of the calves."

-The sentence in lines 38-4 is difficult to understand. Perhaps the authors should take advice from a native speaker.

Response: Thank you for your suggestions. The Introduction, as well as the rest of the article has been rewritten.

-Line 47, biofilms or biofilm?

Response: Thank you for this note, the correct word is biofilm.

Materials and Methods

Section 2.1. Samples, isolation and identification of bacteria

-I suggest rewording. First, specify which animals (species, age) were sampled and how many samples were collected. Time range, place. What types of samples were taken from living and dead animals. The total sum of samples taken and the total sum of animals from which samples were taken (individual animal species). Then the sampling technique and bacterial identification.

Response: Thank you for your feedback. We have restructured this section to begin by outlining the sources of samples, followed by the process of sampling clinical and pathological specimens, and then bacterial cultivation. The final segment focuses on the MALDI-TOF bacterial identification. Additionally, we have simplified the language to maintain clarity while retaining all pertinent information.

-Regarding samples taken from birds. The authors detailed how many samples were taken from hen, rooster and chicken. However, for turkeys, pigeons and ducks, age and gender were not taken into account. I suggest standardizing this data.

Response: Detailed information on gender and age was available only for the chicken, for the other species it was adults of unknown gender.

-Lines 66-67, instead of "Throughout the observed period..." I suggest: "In the period from 2013 to 2017..."

Response: Thank you for this suggestion. It has been rewritten.

-Line 69, Each sample represents one animal. What does it mean? …

Response: The sentence has been reworded: “Only one isolate per animal was included in the study.”

-Sentence, line 73. All types of samples were evaluated in cattle. It is not clear

-Line 74. The statement: Approximately 70% of these samples ... Is imprecise

-Line 78, Almost no dairy calf received mature milk from its own mother. This information is imprecise.

Response to notes for lines 73-78: The entire paragraph has been rewritten.

-Lines 79-80, this sentence is redundant.

Response: This sentence was removed.

Section 2.2. Whole genome sequencing (WGS)

-Lines 117-118, Sentence: "Seven G. anatis isolates from calves and five of our isolates from hens, along with one reference isolate from a hen, were used for genomic comparison (Supplement 1)". If five "our" isolates from chickens were used, whose isolates were from calves?

Response: All sequenced isolates originated from this study, except the CAPM5995, isolated by colleague from our institute in the year 1975. The word “our” has been deleted.

-Table 1 is unnecessary since the authors cite the source in the methodology.

Response: We believe that the information in the table is pretty clear, so we removed duplicate information from the text and kept the Table 1.

Results

-Line 177, "...prevalence within animal species" is redundant. All you need is the percentage in brackets.

-Lines 180-181, what does ".."categories of domestic chicken" mean?

-Lines 186-187. I don't understand this sentence: "While there was only one calf with the autopsy diagnosis of multi-organ infection, G. anatis was isolated from the spleen of two other calves."

-Lines 188-189. The sentence fragment in brackets is unnecessary.

-Lines 190-192. The sentence needs to be reworded because it is difficult to understand.

Response for lines 177-192: The prevalence within animal species has been deleted, the exact data are presented in Table 2. The paragraph was rephrased addressing comments.

-Line 193. (Table 2) show the exact data for diagnoses.

Response: The way in which we refer to tables in the text is common in the journal Pathogens, so we have kept this way.

Table 2. Number and origin of Gallibacterium anatis isolates from veterinary materials in the period

2013 - 2017.

-Table 2 presents data for hens and roosters, and for ducks, turkeys and pigeons without gender division.

Response: The gender division was not known, except for chickens.

-In table 2. Diagnosis rubric - for calves is linguistic error: "eddiarrhea"

Response: Thank you for error detection. It has been corrected.

Section 3.2.2. AMR genotyping

-Line 207, please remove "at all"

Response: It has been deleted.

-Lines 209-210, "In our calf isolates, the occurrence of AMR genes was notably higher" Notably higher than where?

Response: The sentence has been changed: “Calf isolates displayed a significantly higher incidence of AMR genes compared to hen isolates.“

-Paragraph 217-232 should be included in the discussion.

Response: We perceive the comparison of our data with Belgian data as a part of our results, with knowledge that these are distinct strains. From our point of view, by comparing these data, we gained new knowledge about the diversity of different G. anatis populations. The comparison of both groups is stated in the introduction as one of the study's objectives.

-Section 3.2.3. Comparison of hen and calf isolates

-Paragraph 246-252 should be included in the discussion.

Response: The answer is the same as for paragraph 217-232.

The situation is similar to the previous note. We perceive the comparison of hen and calf data as a part of our results, with knowledge that these are distinct strains. From our point of view, by comparing these data, we gained new knowledge about the diversity of different G. anatis populations. Again, the comparison of both groups is stated in the introduction as one of the study's objectives.

-Section 3.2.4. Susceptibility to antimicrobials

-Line 260, the isolated isolates strains

-Line 266, what does "Good results" mean?

-Line 271, what does "weak susceptibility" mean?

Response for the line 260-271: The whole section 3.2.4 has been rewritten.

-Line 274, (Table 4) shows the exact data.

Response: The way in which we refer to tables in the text is common in the journal Pathogens, so we have kept this way.

Discussion

-Lines 291-299. These sentences contain elements of speculation. Information about nutrition and housing conditions, possible contacts with other animal species can be obtained from anamnesis.

Response: Thank you for this note. The sentences were removed.

-Lines 311-314, "The differences in diagnoses between calves and birds may be partly attributed to the numbers of sample types" Please rephrase the sentence because it is difficult to understand.

Response: Here we want to say, that potential bias in comparison of bird and calve samples may be caused by different sample types from calves (in vivo taken rectal swabs and feces samples) and birds (pathological samples), and different numbers of both types of samples. The sentence was changed to :"The differences in diagnoses between calves and birds may be partly attributed to the different sample types.”

-Lines 321-322, I suggest rewriting the sentence: “Screening G. anatis genomes for AMR genes also revealed differences between groups of strains” to: “Screening G. anatis genomes for AMR genes also revealed differences depending on the origin of the strains.”

Response: Thank you very much for suggestion, the sentence was rephrased in the way you suggest.

Conclusions

-Lines 371-373. These sentences are not conclusions. this is a repetition of the purpose and design of the study.

Response: Thank you very much for this note. The Conclusion was rephrased.

Comments on the Quality of English Language

A big problem is the construction of sentences, sometimes introducing ambiguity as to the content. Therefore, I believe that the work should be linguistically proofread. 

Response: The manuscript has undergone significant language editing.

Reviewer 2 Report

Comments and Suggestions for Authors

"Gallibacterium anatis isolated from pathological processes in domestic mammals and birds in the Czech Republic"

The presented paper shows an analysis of G. anatis bacterial isolates by comparing isolates from poultry with sequence data from calves. These bacterial isolates are usually commensals and rarely cause infections. These bacterial species can be found more frequently in poultry, but less frequently in other animal species.

L. 117 Why were so few isolates analyzed? These few studies do not provide a solid database. There were considerably more isolates available.

LL 159-162 As no interpretive criteria are currently available for G. anatis, a classification of the isolate as resistant or susceptible is not possible. How were the MIC examinations performed? A classification into susceptible and resistant is not possible. Please correct and please use current CLSI documents (e.g. APP: VET08 is the current document)

LL 217-218 Why were isolates from bronchopneumonia infection compared with isolates from the gastrointestinal tract?

L 226 It is not surprising that isolates from bronchial pneumonia show macrolide resistance, as treatment with a macrolide is likely. Therefore, an important point is: which animals were pre-treated? This question should be clarified.

Fig 1 Due to the low number of isolates, the phylogenetic tree is not very significant.

L 261 Why was only one isolate tested against cephalexin? Please test the others as well.

L 286-287 G. anatis in calves are probably rather rare cases.

L 324-325 The evident differences are due to the different sites of infection.

L 333-336 The number of isolates examined is very low, only 12 isolates were sequenced, so that the significance is very reduced.

L. 368 G. anatis is certainly only one factor in a multifactorial process in calf diarrhea. The crucial factor is which other pathogenic factors are also present and which pathogens are isolated. The situation is certainly somewhat different in poultry.

The paper would certainly be more informative overall if the sequence results of other isolates, which were also found, were presented.

Comments on the Quality of English Language

The paper should be reviewed by a native speaker

Author Response

Reivewer 2

"Gallibacterium anatis isolated from pathological processes in domestic mammals and birds in the Czech Republic"

The presented paper shows an analysis of G. anatis bacterial isolates by comparing isolates from poultry with sequence data from calves. These bacterial isolates are usually commensals and rarely cause infections. These bacterial species can be found more frequently in poultry, but less frequently in other animal species.

L. 117 Why were so few isolates analyzed? These few studies do not provide a solid database. There were considerably more isolates available.

Response: The project comparing avian and mammalian isolates of G. anatis was ongoing for a longer period of time (2013-2017). The option of whole-genome sequencing was only available towards the end of the project. Due to limited funding for NGS, only some randomly selected isolates from both groups were sequenced. Although the number of sequenced samples was small, the results from their comparison show that the two populations differ. It supports the hypothesis, that bird and calf G. anatis isolates have not the same source.

LL 159-162 As no interpretive criteria are currently available for G. anatis, a classification of the isolate as resistant or susceptible is not possible. How were the MIC examinations performed? A classification into susceptible and resistant is not possible. Please correct and please use current CLSI documents (e.g. APP: VET08 is the current document)

Response: The chapter „2.5. Antimicrobial susceptibility testing” has been rewritten and the use of interpretative criteria has been clarified. Current CLSI and CASFM-VET documents have been included. In our study the MIC examinations were not performed but disk diffusion method was used.

LL 217-218 Why were isolates from bronchopneumonia infection compared with isolates from the gastrointestinal tract?

Response: During the evaluation of the obtained data, the work of the Belgian authors was published (Van Driessche et al. 2020). Due to the obvious differences between our isolates and the Belgian isolates, we thought it appropriate to include the Belgian isolates in the comparison of antibiotic resistance and whole genome sequences and thus gain a deeper insight into the variability of G. anatis isolates. Comparison of the variability of the core genome confirmed significant differences in the genetic relatedness of the isolates.

L 226 It is not surprising that isolates from bronchial pneumonia show macrolide resistance, as treatment with a macrolide is likely. Therefore, an important point is: which animals were pre-treated? This question should be clarified.

Response: We agree that the method of therapy for infections of various organ systems has an effect on the development of pathogen resistance. This is a possible explanation for the differences in antimicrobial resistance and occurrence of AMR genes between our and Belgian isolates. We do not have information on the pretreatment of the investigated animals.

Fig 1 Due to the low number of isolates, the phylogenetic tree is not very significant.

Response: We agree that the inclusion of a larger number of isolates in the whole-genome comparison would refine the information about the variability of the G. anatis population. Sequencing a larger number of our isolates was not feasible, and we utilized all available genomic sequences from Belgian authors. Nevertheless, we believe that Figure 1 adequately illustrates the significant variability within the G. anatis population.

L 261 Why was only one isolate tested against cephalexin? Please test the others as well.

Response: The examination of susceptibility to cephalexin was indicated for only one clinical isolate. Recognizing the limited significance of information based on just one isolate, we have removed the sensitivity data to cephalexin from the manuscript.

L 286-287 G. anatis in calves are probably rather rare cases.

Response: We agree that G. anatis is a rare pathogen in calves, but there is an increasing amount of information regarding its clinical significance.

L 324-325 The evident differences are due to the different sites of infection.

Response: We agree that differences between mammalian and avian isolates can be expected. We omit the claim that it was an interesting observation.

L 333-336 The number of isolates examined is very low, only 12 isolates were sequenced, so that the significance is very reduced.

Response: We agree with you. Please see response to Figure 1 note.

L. 368 G. anatis is certainly only one factor in a multifactorial process in calf diarrhea. The crucial factor is which other pathogenic factors are also present and which pathogens are isolated. The situation is certainly somewhat different in poultry.

Response: Our point of view is very similar to yours. In our study, only isolates where G. anatis was found as dominant pathogen were included.

The paper would certainly be more informative overall if the sequence results of other isolates, which were also found, were presented.

Response: Please see response to Figure 1 note.

Comments on the Quality of English Language

The paper should be reviewed by a native speaker

Response: The manuscript has undergone significant language editing.